# Smad7 Antisense Oligonucleotide in Crohn’s Disease: A Re-Evaluation and Explanation for the Discordant Results of Clinical Trials

**DOI:** 10.3390/pharmaceutics15010095

**Published:** 2022-12-28

**Authors:** Giovanni Monteleone, Carmine Stolfi

**Affiliations:** Department of Systems Medicine, University of Rome “Tor Vergata”, 00133 Rome, Italy

**Keywords:** inflammatory bowel disease, TGF-beta, cytokines, mucosal inflammation

## Abstract

In Crohn’s disease (CD) and ulcerative colitis (UC), the major inflammatory bowel diseases (IBD) in human beings, the tissue-damaging inflammatory response is characterized by elevated levels of Suppressor of Mothers Against Decapentaplegic (Smad)7, an inhibitor of the immunosuppressive cytokine Transforming Growth Factor (TGF)-β1. Consistently, preclinical work in mouse models of IBD-like colitis showed that the knockdown of Smad7 with an antisense oligonucleotide (AS) attenuated the mucosal inflammation, thus paving the way for the development of an AS-containing pharmaceutical compound, named mongersen, for clinical use. The initial phase 1 and phase 2 studies showed that oral administration of mongersen was safe and effective in inducing clinical remission in active CD patients. However, subsequently, a large multicentered, randomized, double-blind, placebo-controlled, phase 3 trial was prematurely discontinued because of an interim analysis showing no effect of mongersen on the activity of CD. In this study we will discuss recent data showing that the majority of the batches of mongersen used in the phase 3 study were chemically different from those used in the previous clinical trials, with some of them being unable to knockdown Smad7 in cultured cells. The accumulating evidence highlights the need to maintain consistent manufacturing requirements for clinical AS, as well as the potential benefits of in vitro bioassays as a part of quality control. New clinical trials evaluating mongersen’s impact on IBD using chemically homogenous batches will be needed to ascertain the therapeutic efficacy of such a drug.

## 1. Introduction

Crohn’s disease (CD) and ulcerative colitis (UC) are chronic, relapsing inflammatory bowel diseases (IBD), which can be associated with the development of local and extra-intestinal complications. The aetiology of CD and UC remains unknown, but in recent decades, a considerable amount of research has contributed to the dissection of the mechanisms leading to the pathogenic processes in these diseases. The current dominating hypothesis is that CD and UC arise because of a complex interaction between several environmental factors and genetic alterations, which eventually triggers an excessive and poorly regulated immune response against luminal antigens [1,2]. These advances have contributed to the development of new and powerful drugs. Indeed, for patients unresponsive or intolerant to mesalamine, corticosteroids, or immunosuppressors, treatment with compounds inhibiting the ongoing mucosal immune response [i.e., anti-cytokines, anti-α4β7 integrin, and Janus Kinase (JAK) inhibitors] induces and maintains remission in CD patients and UC patients [3,4,5]. Nonetheless, half of the IBD patients treated with biologics or small molecules show primary non-response or lose responsiveness over time. Moreover, such drugs can enhance the risk of side effects, thereby leading to the discontinuation of the treatment. This highlights the necessity of further studies to identify additional pathways of IBD-associated tissue damage and/or stratify patients for optimal treatment.

In physiological conditions, the gut mucosa is infiltrated with huge numbers of inflammatory cells, since our gut immune systems are continually exposed to luminal antigens derived from the diet and autologous microflora. Although many of these cells exhibit the features of activated cells and are able to secrete pro-inflammatory molecules, no tissue damage occurs [1]. Indeed, the gut immune homeostasis is tightly controlled by several counter-regulatory mechanisms. One such mechanism involves the activity of suppressive molecules, which are produced by several cell types and inhibit the activation and function of effector cells [6,7]. Support for such concepts comes from pioneering studies in mice, showing that the loss of the interleukin (IL)-10 gene is sufficient to promote chronic intestinal inflammation [8]. In line with this observation, it was then demonstrated that genetic deficiencies in IL-10 and IL-10 receptors cause early-onset and severe IBD in humans, whilst being partially responsive to treatment with corticosteroids or anti-tumor necrosis factor drugs [9]. Similarly, mouse studies showed that the disruption of TGF-β signaling, another regulatory cytokine, led to systemic inflammation, also involving the colon [10]. Crucially, heterozygous mutations in the genes encoding the subunits of the TGF-β receptor cause Loeys–Dietz syndrome and increase the risk of IBD to 10 times that of the general population. In Loeys–Dietz syndrome, IBD develops most frequently in young patients and is marked by a severe clinical course which is poorly responsive to traditional, anti-inflammatory medications [11]. Overall, these findings raise the possibility that defects in counter-regulatory mechanisms not only contribute to the IBD-associated pathological process but also promote the activation of detrimental signals that are not suppressed by the currently available drugs. 

Studies aimed at characterizing the expression and signaling of TGF-β1 during gut inflammation have shown that the cytokine is produced in excess in the inflamed gut mucosa of IBD patients in comparison to the uninflamed mucosa of the same patients [12,13]. Similarly, an elevated production of TGF-β1 was found in the inflamed colon of mice with IBD-like experimental colitis [14]. Nonetheless, both IBD and mouse experimental colitides are marked by a defective activity of TGF-β1, which is due to the enhanced expression of Smad7, an inhibitor of TGF-β1 signaling [12,13,14,15,16]. Consistently, knockdown of Smad7 with a specific antisense oligonucleotide (AS) restored TGF-β1 activity with the downstream effect of suppressing many inflammatory pathways [14,17]. Moreover, studies in experimental models of colitis and in mice with selective over-expression of Smad7 in dendritic cells confirmed the pathogenic role of Smad7 in the gut [18], thus paving the way for the development of a Smad7 AS-containing pharmaceutical compound (named mongersen, formerly GED0301) to use in humans.

## 2. Clinical Trials of Mongersen in IBD

An open-label phase 1 study and an initial, double-blind, placebo-controlled, multicentered, phase 2 study in patients with active CD showed that oral administration of mongersen was safe [19,20,21]. Specifically, in both studies, a clinically active disease was requested at entry and patients had lesions confined to the terminal ileum and/or right colon [22]. The latter inclusion criteria was selected considering the fact that mongersen formulation is protected by an external tablet coating made of pH (6.6–7.2)-dependent methacrylic acid polymers, which allow the active drug to be primarily released in the terminal ileum and right colon. In both studies, mongersen administration was associated with clinical remission in more than fifty percent of the patients. Despite these promising results, a phase 3 study, which was conducted in patients with clinically and endoscopically active disease, was prematurely discontinued, as a futility analysis on 560 patients (421 receiving mongersen and 139 placebo) showed no efficacy of the drug [23]. Although no conclusive explanation was provided, it was claimed that the failure of the phase 3 trial was due to the lack of immunopharmacological effects of the drug, which were over-estimated in phases 1–2, and to the more appropriate inclusion criteria (i.e., documented endoscopic activity of lesions) adopted in the phase 3, which was not used in the previous trials [24]. The latter explanation seems to be reductive, as another phase 2, non–placebo controlled study, in 63 CD patients with endoscopic evidence of active inflammation in the gut, demonstrated both decreased clinical and endoscopic activity [25]. It is thus plausible that the conflicting results generated in those studies could rely on additional factors, which could have influenced the efficacy of mongersen in the phase 3 trial.

## 3. Loss in Bioactivity for Batches of Mongersen Used in Trials

The AS contained in mongersen is a single-stranded, synthetic oligonucleotide that hybridizes to the region 107–128 of the human Smad7 mRNA in a sequence-specific manner, thereby triggering RNase H1 activity, mRNA degradation, and, consequently, downregulation of the Smad7 protein. The AS is chemically modified since phosphorothioate (PS) is used as a substitute for the phosphodiester (PO) linkages between nucleotide bases. Such a chemical modification is known to improve the metabolic stability and cellular uptake of AS without compromising their affinity for target mRNA or RNase H1 activity [26]. However, PS substitution converts the non-chiral PO linkage into a chiral PS center, having two distinct stereochemical configurations, named *S*p and *R*p. Such a modification leads to the formation of 2^n^ diastereomers, with *n* being the number of PS linkages present in the PS oligonucleotide. As there is no practical means to neither separate the individual stereoisomers generated during the synthesis of PS-modified oligonucleotides nor synthesize stereochemically pure oligonucleotides, all PS-modified AS developed for clinical use are a mixture of diastereomers, which may have distinct behaviors in vitro and in vivo [27]. 

Several batches of mongersen were manufactured and used in the clinical trials. The dissolution and purity of all these batches were similar, but recent studies showed that the stereochemistry was not homogenous among the different batches and the small batches made for the phase 1 and 2 studies had stereochemistry, which differed from that seen in the large batches made for the phase 3 study [28]. Interestingly, in vitro bioassays documented significant differences in terms of Smad7 knockdown. Specifically, the batches used in the phase 1 and 2 studies, and some of those used in the phase 3 study, inhibited Smad7 RNA and protein in colon cancer cell lines, whereas the majority of batches used in the phase 3 had minimal or no inhibitory effect [28]. Moreover, analysis of the batches by solution phosphorus-31 nuclear magnetic resonance (^31^P-NMR) spectroscopy, which can assess the chemical environments of the phosphorus atoms, showed that each mongersen preparation had a distinct ^31^P-NMR profile, indicating that the batches that were used in the clinical trials had a distinct PS chirality. By principal component analysis, we were also able to identify several clusters of the batches with similar ^31^P-NMR spectra, and preparations with the same ^31^P-NMR spectrum profile which had similar in vitro activity, as shown by their inhibitory effect of Smad7 in cultured cancer cell lines [28]. Further analysis of the patients who received mongersen during the phase 3 clinical trial showed that those treated with batches exhibiting the most powerful in vitro activity had the greatest reductions in clinical activity of the disease [28]. Consistent with this are the results of a more recent, phase 2, open-label, study, in which 18 clinically and endoscopically active CD patients received mongersen 160 mg/day for 12 weeks. This treatment was associated with clinical benefit in more than fifty percent of the patients. Furthermore, the pharmacological batch of mongersen used in this study inhibited Smad7 expression in cultured cancer cells [29].

## 4. Conclusions

Overall, data emerging from pre-clinical work indicates that IBD is marked by a high expression of Smad7 in the inflamed gut, which results in defective anti-inflammatory activity of TGF-β1 and amplification of the ongoing mucosal inflammation. Nonetheless, studies in CD patients have provided conflicting results about the effect of Smad7 knockdown on the course of the disease. No conclusive evidence has been produced, as of yet, to explain such discrepancies, but recent studies support the hypothesis that differences in the diastereomeric content of the various batches of mongersen developed during the whole development program could have contributed to the generation of different results as seen in the clinical trials, thereby explaining the failure of the phase 3 trial. This is in line with the principle that patients treated with PS-modified AS received a mixture of thousands of diastereoisomers bearing distinct three-dimensional structures and pharmaceutical properties. The available evidence also seems to indicate that the small batches developed to conduct the small phase 1 and phase 2 trials had similar PS chirality, which differed from that seen in the majority of the batches used in the phase 3 clinical trials. Clearly, additional work is warranted to further address this issue and ascertain whether specific changes in the manufacturing protocols can reduce the diastereomeric complexity. In this context, it is noteworthy that recent studies have shown that the application of symmetrical non-bridging PS linkages, in the context of stereodefined AS, reduces the chiral complexity, thus resulting in the generation of single molecules [30].

While these steps are mandatory before moving into clinics and re-evaluating the therapeutic efficacy of mongersen in IBD, additional experimentation is needed to further examine the contribution of Smad7 in the IBD-associated inflammatory response. Indeed, recent studies have shown that, besides its action as an antagonist of TGF-β1 signaling, Smad7 can interact with a variety of nuclear proteins [31,32], thereby controlling multiple pathways, which could be relevant for IBD.

## Data Availability

Data describing the chemical properties of the mongersen batches are available in PubMed and can be accessed via the following DOI link: 10.1089/nat.2021.0089.

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
