# Peer review of "Smad7 Antisense Oligonucleotide in Crohn’s Disease: A Re-Evaluation and Explanation for the Discordant Results of Clinical Trials"

_pharmaceutics, 2022, doi:10.3390/pharmaceutics15010095_

Round 1

Reviewer 1 Report

Recently, inflammatory bowel diseases (IBD) currently represent a hot research field. There are a lot of papers about this research field. However, this field lacks a short communication for Smad7 antisense oligonucleotide in Crohn's disease. In this manuscript, the authors finished one short communication with good quality and comprehensive summary. This manuscript summarized recent data showing that most of the batches of Mongersen used the phase 3 study were chemically different from those used in the previous clinical trials with some them being unable to knockdown Smad7 in cultured cells. This work and some of the conclusions presented very interesting. In my opinion, it can be published in Frontiers in Immunology. This has the potential to be a really good review, however, there are still many deficiencies in this work that need to be modified. Prior to be published in Pharmaceutics, some minor corrections are necessary:

-The manuscript needs another revision to check for spacing (spaces after words and before the citations are often missing).

- All acronyms must be introduced in their first appearance. There isn’t an abbreviation list. Some undefined abbreviations in the manuscript are confusing to the readers.

Author Response

Reviewer #1

We thank the Reviewer for his/her positive evaluation.

In response to the specific points raised by this Reviewer:

1. The manuscript needs another revision to check for spacing (spaces after words and before the citations are often missing).

A. We carefully checked the manuscript and fixed the indicated issues

2. All acronyms must be introduced in their first appearance. There isn’t an abbreviation list. Some undefined abbreviations in the manuscript are confusing to the readers.

A. All the acronyms have been spelt out at their first mention.

Reviewer 2 Report

The manuscript is well-written and well-thought. I totally agree with the authors that different batch preparations will yield different results.

It is essential to keep the protocols and manufacturing processing the same while testing the anti-SMAD7 drug Mongersen in all three phases of clinical trials. It is highly likely that different manufacturing processes produced heterogeneity in Mongersen drug as authors have clearly pointed out, so It makes complete sense that different results had been seen in the phase three clinical trial.

Author Response

Reviewer #2:

We thank the Reviewer for his/her positive evaluation.

The manuscript is well-written and well-thought. I totally agree with the authors that different batch preparations will yield different results. It is essential to keep the protocols and manufacturing processing the same while testing the anti-SMAD7 drug Mongersen in all three phases of clinical trials. It is highly likely that different manufacturing processes produced heterogeneity in Mongersen drug as authors have clearly pointed out, so It makes complete sense that different results had been seen in the phase three clinical trial.

We appreciate the positive feedback from Reviewer 2.

Reviewer 3 Report

The manuscript addresses an important point about maintaining quality control in batches of drug products (Mongersen) before it is utilized in clinical trials to attain conclusions. Although one author has personal interest in the matter it will still be something of interest to the scientific audience in the filed. 

1. Please provide a visual representation of the findings to accompany the text, this will grasp the readers attention to this issue more promptly

2. How can the issue of batch variation be addressed? Please provide or discuss a solution

3. Can the authors cite more references which recognizes work from groups other than the first author?

Author Response

Reviewer #3

We thank the Reviewer for the positive comments and helpful suggestions.

In response to the specific points raised by this Reviewer:

  1. Please provide a visual representation of the findings to accompany the text, this will grasp the readers attention to this issue more promptly

A. We thank the Reviewer for the helpful suggestion. A graphical abstract summarizing the key points of the paper has been included in the revised files.

  1. How can the issue of batch variation be addressed? Please provide or discuss a solution

A. We made clear in the conclusions that a possible solution to address the issue of batch variation could be the application of symmetrical non-bridging PS linkages in the context of stereodefined AS. This process may reduce the chiral complexity thus resulting in the generation of single molecules. At the same time, we pointed out that additional work is warranted to further address this concern and ascertain whether specific changes in the manufacturing protocols could reduce the diasteromeric complexity.

  1. Can the authors cite more references which recognizes work from groups other than the first author?

 A. To address this issue we have removed the old ref. 12 (Monteleone et al, JBC 2004) and included more references recognizing work from other groups (see new refs 2, 5, 7, 15, 16, and 32)